# Altered Cerebro-Cerebellar Effective Connectivity in New-Onset Juvenile Myoclonic Epilepsy

**DOI:** 10.3390/brainsci12121658

**Published:** 2022-12-03

**Authors:** Laiyang Ma, Guangyao Liu, Pengfei Zhang, Jun Wang, Wenjing Huang, Yanli Jiang, Yu Zheng, Na Han, Zhe Zhang, Jing Zhang

**Affiliations:** 1Department of Magnetic Resonance, Lanzhou University Second Hospital, Lanzhou 730030, China; 2Second Clinical School, Lanzhou University, Lanzhou 730030, China; 3Gansu Province Clinical Research Center for Functional and Molecular Imaging, Lanzhou 730030, China; 4School of Physics, Hangzhou Normal University, Hangzhou 311121, China; 5Institute of Brain Science, Hangzhou Normal University, Hangzhou 311121, China

**Keywords:** degree centrality, effective connectivity, cerebellar, juvenile myoclonic epilepsy

## Abstract

(1) Objective: Resting-state fMRI studies have indicated that juvenile myoclonic epilepsy (JME) could cause widespread functional connectivity disruptions between the cerebrum and cerebellum. However, the directed influences or effective connectivities (ECs) between these brain regions are poorly understood. In the current study, we aimed to evaluate the ECs between the cerebrum and cerebellum in patients with new-onset JME. (2) Methods: Thirty-four new-onset JME patients and thirty-four age-, sex-, and education-matched healthy controls (HCs) were included in this study. We compared the degree centrality (DC) between the two groups to identify intergroup differences in whole-brain functional connectivity. Then, we used a Granger causality analysis (GCA) to explore JME-caused changes in EC between cerebrum regions and cerebellum regions. Furthermore, we applied a correlation analysis to identify associations between aberrant EC and disease severity in patients with JME. (3) Results: Compared to HCs, patients with JME showed significantly increased DC in the left cerebellum posterior lobe (CePL.L), the right inferior temporal gyrus (ITG.R) and the right superior frontal gyrus (SFG.R), and decreased DC in the left inferior frontal gyrus (IFG.L) and the left superior temporal gyrus (STG.L). The patients also showed unidirectionally increased ECs from cerebellum regions to the cerebrum regions, including from the CePL.L to the right precuneus (PreCU.R), from the left cerebellum anterior lobe (CeAL.L) to the ITG.R, from the right cerebellum posterior lobe (CePL.R) to the IFG.L, and from the left inferior semi-lunar lobule of the cerebellum (CeISL.L) to the SFG.R. Additionally, the EC from the CeISL.L to the SFG.R was negatively correlated with the disease severity. (4) Conclusions: JME patients showed unidirectional EC disruptions from the cerebellum to the cerebrum, and the negative correlation between EC and disease severity provides a new perspective for understanding the cerebro-cerebellar neural circuit mechanisms in JME.

## 1. Introduction

Juvenile myoclonic epilepsy (JME) is a subtype of idiopathic systemic epilepsy (IGE) that occurs between 12 and 18 years of age and accounts for 5~10% of all epilepsy [1]. The main clinical manifestations of JME include myoclonic spasms, generalized tonic-clonic seizures, and absence seizures. As a conventional MRI shows no organic lesions in this condition, the final diagnosis is based on the typical history and electroencephalogram (EEG) findings showing an abnormal change (typical high amplitude of EEG features of a 3~6 Hz generalized spike-wave or polyspike-wave discharge [GSWD]) [2,3]. As the disease progresses, patients develop long-term neuropsychological and cognitive impairments, such as memory and attention deficits, and even executive dysfunction [4]. So far, the etiology of JME, the neurophysiological basis of the disease, and the specific locations of the abnormal discharges remain unknown.

Resting state functional connectivity (rsFC) has been widely used in the study of epilepsy. Vollmar et al. demonstrated that when JME patients perform working memory tasks, the functional connectivity (FC) of the motor system is enhanced, and the auxiliary motor area may play an important role in connecting the cognitive network and the motor system [5]. Abnormalities in the thalamus-prefrontal cortex network may be related to the transmission of generalized spikes in JME [6,7], and the differences in network connections with the salience network (SN) and default mode network (DMN) may be related to emotional and cognitive defects in JME [8,9].

Additionally, previous studies have shown cerebellar dysfunction in patients with JME, suggesting that the cerebellum plays a key role in epileptic seizures [10]. Marcián et al. [11] identified cerebellar volume changed in patients with temporal lobe epilepsy. Some scholars have reported abnormalities in static FC, language and cognitive networks between the cerebellum and the brain in epilepsy patients through resting-state FC anal-yses [12,13]. JME patients also showed abnormal cerebellar structure and function, and FC changes have been found in motor- and cognition-related areas [14,15]. These findings suggest that epilepsy is mediated by a complex network of changes and that abnormalities in the cerebro-cerebellar microstructural and FC may be an important feature of JME.

Traditional FC studies mainly focused on the FC of seed points or independent component analysis (ICA), and few studies have explored changes in topological properties. DC approach using a voxel-level graph analysis method can depict the nodes in the network center for quantitative analysis without requiring region of interest (ROI) selection. The number of FCs can, to a certain extent, reflect the complex brain network traffic characteristics, compensating for the limitations of traditional FC research. This approach has been widely used in studies of the effects of disease on brain network attributes [16,17].

While FC only measures instantaneous (that is, zero time lag) temporal correlations between spatially different brain regions, and is therefore undirected, effective connectivity (EC) measures neural activity in the causal (and thus directional) effects of one region on another [17,18]. Thus, an EC analysis of JME can characterize the information flow between brain regions and therefore may provide a richer explanation of the neuropathology of JME [19]. Moreover, our previous study made the first attempt to characterize the dynamic effective connectivity (DEC) in the DMN of patients with JME, demonstrating that the DEC between brain networks is very important in JME [20]. Epilepsy is regarded as a disruption of functional networks [21], and an increasing number of studies have suggested that cerebro-cerebellar interactions are important for the induction of JME [22,23,24,25].

Therefore, this study aimed to explore the characteristics of core network EC in newly diagnosed JME. To characterize the brain functional network architecture in JME, we sought to identify brain regions with different DC values in JME in comparison with healthy controls (HCs) and used EC measurements to understand the directional aspect of these alterations. We hypothesize that the cerebro-cerebellar circuit network in JME patients plays an important role in the mechanism of epilepsy.

## 2. Materials and Methods

### 2.1. Participants

Thirty -four participants with newly diagnosed JME were recruited from the Epilepsy Center of Lanzhou University Second Hospital. while thirty-four age-, sex-, and education- matched HCs were recruited from the hospital through advertisements (see Table 1 for details). The diagnosis criteria of JME was based on the classification criteria of the International League against Epilepsy in 2001 [26]. The specific inclusion criteria of JME included: normal routine MRI scans and 3~6 Hz generalized spike-wave or polyspike-wave discharges (GSWDs) on routine scalp EEG. The seizure severity of each patient was measured using the National Hospital Seizure Severity Scale (NHS3) [27]. The exclusion criteria for the patient group were as follows: (1) a history of any forms of antiepileptic treatment, (2) other neurological or psychiatric illness, (3) other developmental disabilities, such as autism or intellectual impairment, (4) acute physical illnesses that could affect the scans, (5) claustrophobia or metal implantation that was not suitable for MR scanning; and (6) alcohol, tea and caffeine intake. In addition, HCs were screened for neurological or psychiatric disorders.

All participants were informed of the purpose and content of the experiment and voluntarily provided informed consent. This study was approved by the Medical Ethics Committee of Lanzhou University Second Hospital.

### 2.2. Magnetic Resonance Imaging Acquisition and Data Preprocessing

All resting-state fMRI data were collected using a Siemens Verio 3.0 T scanner at Lanzhou University Second Hospital. In the process of data collection, participants were required to wear special mute earphones and earplugs, and their heads were fixed with sponge pads. Participants were instructed to remain awake, stay relaxed with their eyes closed, and not perform any action or intentional thinking. To mitigate the possible influence of residual EEG signals from seizures on the resting-state fMRI data, we confirmed that the JME patients had not experienced any seizures within 3 days prior to scanning. A gradient-echo echo-planar imaging (GRE-EPI) sequence was used for fMRI collection, and the collection parameters were as follows: repetition time (TR) = 2000 ms, echo time (TE) = 30 ms, flip angle = 90°, slice thickness = 4 mm, in-plane matrix resolution = 64 × 64, field of view (FOV) = 240 × 240 mm^2^, slices = 33, and total scan volume = 200. For anatomical localization and normalization, high-resolution 3D T1-weighted images were obtained using a magnetization-prepared rapid gradient-echo (MPRAGE) sequence (TR = 1900 ms, TE = 2.99 ms, flip angle = 90°, slice thickness = 0.9 mm, acquisition matrix = 256 × 256, FOV = 230 × 230 mm^2^, in-plane resolution = 0.9 × 0.9 mm^2^, slices = 192).

DPARSF software (http://rfmri.org/DPARSF (accessed on 21 December 2021)) was used for the batch preprocessing of fMRI data and the subsequent related analysis [28]. The data preprocessing protocol was as follows: (1) The first 10 time points of the MR signal were excluded to remove the noise generated by the participant’s discomfort at the beginning and the magnetic saturation effect of the machine, ensuring that the MR signal was as stable as possible and leaving 190 time points of the signal for further analysis. (2) Slice timing was performed such that the signals collected at different time points were corrected to the same time point. (3) Head movement correction was performed to reduce the noise generated by participants’ involuntary movement during the process of data collection. The test standard was to exclude the data showing translation > 2 mm and rotation > 2°. We also calculated the average frame-wise displacement (FD) for each participant on the basis of the adjustment parameters, after excluding participants with an average FD greater than 0.2 mm. (4)For spatial standardization, the original space of the participants was estimated to the standard space, such that all functional images were normalized to theMontreal Neurological Institute (MNI) standard space template provided by SPM8, which was used to locate the brain activation region, and each voxel was resampled to 3 mm × 3 mm × 3 mm. (5) The standardized image was smoothed using a 6 mm full width at half maximum (FWHM). (6) Linear drift and regression were removed to remove covariables (including six head parameters, the global signal, the white matter signal, and the cerebrospinal fluid signal), and the linear trend accumulated due to the long collection time was removed.

### 2.3. Degree Centrality Analysis

REST V 1.8 software [29] was used for data analysis with a package (http://www.restfmri.net (accessed on 21 December 2021)) for DC analysis and the calculations were based on the whole-brain DC of the voxels (threshold, 0.25). Briefly, each voxel was regarded as a node, and the linear correlation (Pearson correlation) of the neural activity signals between voxels was regarded as an edge. For each participant, the correlation between any pair of voxels was calculated in the default brain template, and the domain value (r > 0.25) was then set for each group of correlations. Subsequently, the whole-brain functional network was constructed by thresholding each correlation at r > 0.25 to obtain the DC value at each individual level, which was then divided by the mean of the whole-brain DC value to obtain the standardized DC value. To improve data normality, the correlation coefficient was converted to Z values by a Fisher Z transformation, as previously described [30,31]. Then, the centrality distribution map of the Zvalue degree of each participant’s brain FC group was obtained. The DC plots were eventually smoothed with Gaussian smoothing kernel 6-mm FWHM for further statistical analysis.

Voxel-based DC can measure the voxel-level topology of the whole-brain functional connectivity. The DC represents the number of direct connections (or significant suprathreshold correlation weights) of a given voxel in the voxel-linker and is a measure of the importance of individual nodes. It also reflects the “hub” property of brain functional networks, that is, the ability to communicate network information [32].

### 2.4. Effective Connectivity Analysis

Using a DC approach, we were able to show that the brain regions such as the CePL.L, ITG.R, IFG.L, and SFG.R were regions of special functional importance showing changed FC in JME patients, especially in the cerebellum. To further investigate the influence of directionality, we applied a GCA to evaluate changes in EC. Based on the results of the DC analysis, we selected regions of interest (ROIs) in brain regions showing significant differences between JME patients and HCs (CePL.L, ITG.R, IFG.L, and SFG.R). All ROI coordinates were in MNI space (Table 2 shows the specific coordinates). EC was analyzed using REST-GCA in the REST toolbox [33].

EC is a measure of the information transmission pattern between brain regions, and it can establish the causal relationships between neurons, indicate the direct or indirect influence between nervous systems, and reveal the dynamic processes underlying neural activity; consequently, EC-based findings are more consistent with the real brain function mechanism. GCA, as an effective connectivity analysis method, can model the forms of interactions between significantly different brain regions [34].

### 2.5. Statistical Analysis

Statistical analyses were performed using SPSS 17.0 statistical analysis software, and measurement data were expressed as mean ± standard deviations ($\overset{-}{x}$ ± S). The Kolmogorov–Smirnov method was used to test the normality of the data for age, education level, etc. REST software was used to analyze the DC and EC maps after Fisher Z transformation. All measurement data showed a normal distribution. Two independent-samples *t*-tests were used for comparisons between the two groups. A X^2^ test was used for count data comparisons between the two groups, and the significance level was set at *p* < 0.05. The correlation between aberrant EC and the NHS3 score was analyzed in the JME group. AlphaSim (*p* < 0.05) was used to correct the results and record the activated brain regions.

## 3. Results

### 3.1. General Data

The two groups showed no statistically significant differences in sex, age, or education level (*p* > 0.05, Table 1).

### 3.2. Degree Centrality Analysis

Compared to HCs, patients with JME showed significantly increased DC in the CePL.L, ITG.R, and SFG.R and decreased DC in the IFG.L and STG.L (all *p* < 0.05, AlphaSim-corrected; Figure 1 and Table 2).

### 3.3. Effective Connectivity Analysis

Compared to HCs, patients with JME demonstrated increased ECs from cerebellum regions to the cerebrum regions, including from the CePL.L to the PreCU.R, from the CeAL.L to the ITG.R, from the CePL.R to the IFG.L, and from the CeISL.L to the SFG.R (AlphaSim corrected, *p* < 0.05; Figure 2 and Table 3).

### 3.4. Correlation Results

There were no significant correlations between the DC of the differences in brain areas and the severity of epilepsy. However, the EC from the CeISL.L to the SFG.R was negatively correlated with the NHS3 score (*r* = 0.35, *p* = 0.042, FDR corrected, Figure 3).

## 4. Discussion

Modern brain network studies have showed that diseases associated with changes in brain network function typically do not show uniform whole-brain involvement, but instead affect some important core nodes [35].The core nodes in information processing of brain network function are very important structures. The lesion-induced changes in the functioning of the core nodes can cause a series of brain functional network changes [36]. DC usually reflects the core nodes of the brain network, while EC can reveal the abnormalities in directed connectivity in important brain regions.

In this study, using DC analysis, we found significantly aberrant network centrality within the CePL.L, ITG.R, SFG.R, IFG.L and STG.L in patients with JME. Then, using a GCA algorithm, we considered the regions showing differences in DC values as seeds to examine their EC with the whole brain, and the JME patients showed unidirectionally increased ECs from the cerebellum regions to the cerebrum regions, including from the CePL.L to the PreCU.R, from the CeAL.L to the ITG.R, from the CePL.R to the IFG.L, and from the CeISL.L to the SFG.R. Additionally, the EC from the CeISL.L to the SFG.R was negatively correlated with the NHS3 score. These results indicate that abnormalities of cerebro-cerebellar functional centrality and directed connectivity patterns may be important characteristics of JME.

### 4.1. Analyses of Differences in DC Values between Groups

#### 4.1.1. The Encephalic Region and the Significance of Increased DC Values

In the present study, we observed increased DCs in some brain regions, including the CePL.L, ITG.R and SFG.R. The posterolateral cerebellar hemisphere is involved in cognitive behavioral activities such as attention, working memory and visual memory [37]. Cerebellar structure and function may be affected by abnormal epileptic discharge or is involved in the neural network of epilepsy [38,39]. Meanwhile, the abnormal FCs between the posterolateral cerebellum with the thalamus and the auxiliary motion zone (supplementary motor area, SMA) may be related to the movement symptoms in patients with JME [22,40,41]. Our results suggest that the cerebellum plays a prominent role among the brain network structures affected by JME. The findings of some previous FC studies [14,42], which also showed extensive FC enhancement in the cerebellum, were consistent with the DC enhancement reported in the present study. Kros et al. [43] found that the regulation of neuronal firing in the cerebellar nucleus (CN) could affect absence seizure by interfering with cortical oscillations through the innervation of various thalamic nuclei. Sato et al. [44] reported cerebellar hyperperfusion in patients with seizures and suggested that this mechanism may be related to “reactive plasticity“ [45]. Thus, it may be helpful to scientifically explain the phenomenon of the overload or loss of core nodes when processing information [36].

The ITG is known to be involved in high-level cognitive functions, including visual and language comprehensions and emotion regulation [46].The ITG is connected to other cortical regions through u-fiber, the superior longitudinal fasciculus, the arcuate fasciculus and the uncinate fasciculus, including the frontal lobe, parietal lobe, superior temporal gyrus limbic lobe, etc. [46]. Therefore, the ITG is a critical region in the global brain network, and abnormal activation of the ITG could impair temporal lobe function. Shi et al. [13] found that changes in functional homotopy and connectivity in the ITG were associated with temporal lobe epilepsy. The activation of the ITG has also been reported to be associated with sadness [47]. Therefore, as mentioned in previous articles, an abnormal ITG could be used as a neuroimaging marker for the diagnosis and prognosis of brain disorder [13,48].

The SFG is now known to be involved in several functional networks related to motor activity, working memory, resting-state regulation, and cognitive control [49,50,51,52], and is an important part of the prefrontal lobe and a critical part of the frontoparietal system, which is essential for consciousness processing [53]. A variety of epilepsy models based on neuroimaging involve different regions of the prefrontal lobe. Stéphanie et al. [53] found that, although most prefrontal seizures manifest as a network of several anatomically distinct structures, the dorsolateral prefrontal lobe is a relatively defined subgroup. Doucet et al. [54] suggested that the prefrontal cortex (PFC) is an occult regulator of epileptogenesis. While JME is a complex disease, abnormalities in the thalamo-frontal cortex pathway have been confirmed in most studies [55]. These results are consistent with our findings. Therefore, the PFC, especially the SFG, may be one of the important nodes involved in epilepsy.

In summary, the regions showing increased DC in this study covered the cerebrum and cerebellum, suggesting that these regions had increased functional connectivities with other brain regions and that their centrality and importance are more than normal, potentially reflecting a compensatory mechanism to maintain cognitive needs in the brain in JME patients [56]. Another possibility is that neuronal hyperactivity in these regions may counteract or reduce structural brain damage [36].

#### 4.1.2. The Encephalic Region and the Significance of Decreased DC Values

Decreased DC values in JME patients were mainly observed in the IFG.L and STG.L. In humans, the IFG, a major part of the cortex involved in multiple white matter pathways, is a gyral complex that lies ventral to the inferior frontal sulcus, anterior to the precentral sulcus, and superior to the lateral (Sylvian) fissure (LF) [57]. The IFG is involved in several networks related to language, mutual sensory, and affective functions [58]. It is also an integral part of the central control network(CEN) [59]. Cristiano et al. [60] identified the brain network executing attention in the IFG by electrocorticographic (ECoG) stimulation. Therefore, given the decreased DC values observed in the IFG.L, we speculate that the reduction in the relevant functional activity in this brain region may reflect the impaired connectivity of its neural network. This is consistent with the study by Takuro et al. [61]. Simultaneously, the increased EC from the CePL.R to the IFG.L indicates that the CePL.R can promote information transmission from the IFG.L. Alternatively, this finding may indicate some form of compensation for the hypofunction of the inferior frontal region. A recent study in mice showed that the continuous expression of information in the frontal cortex is dependent on the cerebellum [62], which may explain our results.

The STG, which is located on Wernicke’s area, belongs to the limbic system and acts as a critical hub in the auditory network, verbal processing and episodic memory [63,64]. Kim et al. revealed that the left STG is involved in receptive language [65]. Abnormal connectivity of the STG may be related to the sensory and visual auras that appear before seizures [13]. The results of our study showed decreased DC values for the left STG, potentially indicating less reliance on receptive language centers in verbal memory processing. Among the studied brain networks, the limbic network was found to be more affected by temporal lobe epilepsy compared to other networks [66]. Previous studies using graph theory have demonstrated that hypoconnectivity was confirmed in the STG.L and right caudate [66]. This is consistent with the findings of our study. Therefore, the decreased DC in some regions in this study may be explained by the decreased functional activity associated with these brain regions and the impairment in the role of these functional brain regions in facilitating neural network connections.

### 4.2. Effective Connectivity Analysis

In this study, GCA was used to investigate the changes in EC among the core brain networks based on the seed points from brain regions with different DC values. Previous studies have demonstrated different afferent and efferent pathway alterations between the cerebellum and different cerebral cortices in patients with epilepsy [23,67]. However, the EC between the cerebrum and cerebellum in JME patients remains unclear. Our results showed unidirectionally increased ECs from cerebellum regions to cerebrum regions, including from the CePL.L to the PreCU.R, from the CeAL.L to the ITG.R, from the CePL.R to the IFG.L, and from the CeISL.L to the SFG.R. In traditional anatomy, the cerebellum is divided into three regions: the anterior cerebellar lobe, the posterior cerebellar lobe, and the flocculonodular lobe, and there are functional differences between the anterior and posterior lobes of the cerebellum. The anterior cerebellum mainly regulates motor function, while the posterior cerebellum is responsible for cognitive regulation [68]. Structurally, the anterior cerebellar lobe receives spinal afferents through the spinocerebellar tract [69] and is interconnected with the motor cortex through feedforward motor cortical-pontine projections [68,70]. Functionally, the brain networks map onto the cerebellum with the specificity of a topographic map. The motor network is localized to the sensorimotor portion of the anterior cerebellum and lobule VIII [68]. These findings indicate that the cerebellum communicates with multiple cortical regions through different pathways.

As the core node of the DMN, the precuneus showed the most active metabolism in the resting state of the healthy human brain. It mainly plays a leading role in episodic memory and also plays a key role in default mode processing, self-awareness, etc. [71,72]. JME has been repeatedly shown to be associated with brain dysconnectivity in the DMN [14,20]. Qin et al. [73], using synchronous EEG and fMRI methods, found that the blood oxygen level dependent (BOLD) activation areas related to high EEG network variation in JME patients were mainly located in the thalamus, cerebellum, and precuneus. Some studies have shown that, in comparison with HCs, patients with temporal lobe epilepsy showed abnormal EC between the precuneus and the supramarginal gyrus [74,75]. Vaudano et al. [76] applied GCA and dynamic causal modeling (DCM) to show that the characteristic generalized spike-wave (GSW) initiation and persistence network in JME involved the frontal lobe, precuneus, and thalamus and that the precuneus played an important role in triggering the spread of GSW in the corticothalamic network.

An effective connectivity analysis was used to analyze the interactions between statistically significant brain regions through a data-driven analysis to speculate on the structural forms of potential neural connections. EC analysis has been widely used in recent brain network research studies [77,78,79]. Sokolov et al. [80] used a dynamic causal model to show interactive connections between the left lateral cerebellum and the right posterior superior temporal sulcus, which were involved in a wide range of social cognitive functions. Our previous study found alterations in the dynamic EC of the DMN in JME patients [20]. Therefore, analysis of the brain disease network through EC evaluations is a feasible approach. In this study, four EC relationships from the cerebellum to the cerebrum were all positively activated by GCA. Therefore, these four connections are all promoting, that is, the information transmission between brain regions is promoting. The activation pattern from the cerebellum to the cerebrum through ECs may constitute a local pathway with an abnormal connection pattern, and the abnormality of this local pathway may lead to abnormal activity in the brain area near the pathway, which may further expand to a larger area, and may even affect the whole brain. Abbasi B et al. [81] found increased FC between cerebellar anterior lobe I and the inferior parietal lobules, which affected the executive network, in patients with right temporal lobe epilepsy. Gotman et al. [82] found that bilateral cerebellar regions were activated during discharge in GTCS patients, which confirmed that the cerebellum was involved in the generation and propagation of epileptic activity, indicating that the cerebellum may play a certain role in the regulation of the discharge process [43]. In brain diseases, there are many local pathways composed of abnormal connectivity patterns, for example, the cortico-striato-thalamo-cerebellar networks of generalized tonic–clonic seizures (GTCS) [83]. Therefore, damage to some brain regions not only affects the normal function of each brain region but also affects normal FC between the brain regions, resulting in abnormal local pathways. In this study, abnormal ECs may be the key to the local pathway abnormalities of JME.

### 4.3. Correlation Analysis

We also found that the EC from the CeISL.L to the SFG.R was negatively correlated with the NHS3 score in patients with JME, potentially suggesting that the higher ECs correspond to lower disease severity. The most likely route for the cerebellar contribution to cognition is through interactions with the neocortex [84]. Danielson et al. considered that the cerebellum may play an essential role in the network inhibition mechanism [85], while Schmahmann et al. [86] proposed that the cerebellum is a functional regulator. Our previous study also showed that the DEC in the DMN in JME is consistent with observations of associations between seizure symptoms [20]. Taken together, the associations established in the present study suggest that abnormalities in this circuit from the posterior cerebellum to the superior frontal gyrus may be involved in the pathogenesis of JME.

#### Limitations

Our study had some limitations that require consideration. First, this study was based on the effective connectivity of resting-state fMRI. Although Granger analysis based on model algorithms has been proven to be robust [87], future work should incorporate the findings of other multimodal fMRI techniques and employ evaluations using different methods such as dynamic causality. Second, our experiment was designed as a cross-sectional study, making it difficult to obtain direct causal inferences about the relationship among brain functional network structures in JME patients. Further longitudinal studies with more participants are needed to address this issue. Third, our sample size was limited, and researchers should aim to evaluate these findings in larger study populations.

## 5. Conclusions

Our results showed that the core network connectivity in JME patients was abnormal, especially in the anterior and posterior cerebellar lobes. Moreover, patients with JME exhibited abnormal unidirectional effective connectivity from the cerebellum to the cerebrum. The effective connectivity between special nodes was related to the clinical severity, especially in the CeISL.L and SFG.R circuits. These findings suggest that the cerebellum and its wider epileptic network of effective interactions with the cerebrum may underlie the pathological mechanism of JME.

## Figures and Tables

**Figure 1 brainsci-12-01658-f001:**
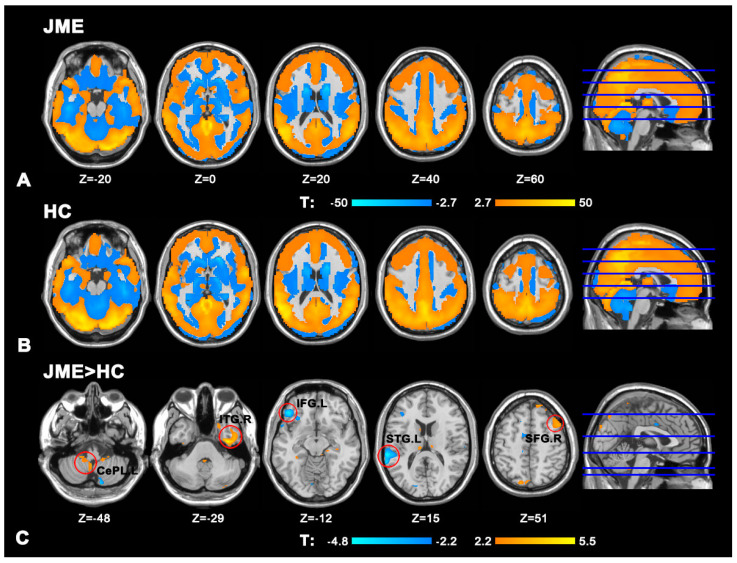
(**A**) Spatial distribution of the degree centrality (DC) in the juvenile myoclonic epilepsy (JME) patients. (**B**) Spatial distribution of the DC in healthy controls (HCs). (**C**) Significantly increased DC within the left cerebellum posterior lobe (CePL.L), the right inferior temporal gyrus (ITG.R), and the right superior frontal gyrus (SFG.R) between the JME patients and HCs, and decreased DC in the left inferior frontal gyrus (IFG.L) and the left superior temporal gyrus (STG.L).

**Figure 2 brainsci-12-01658-f002:**
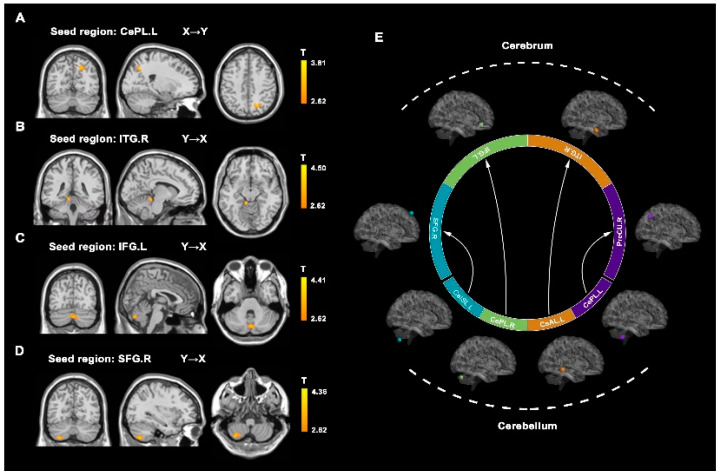
Aberrant effective connectivity in juvenile myoclonic epilepsy (JME) patients. (**A**) Increased effective connectivity from the left cerebellum posterior lobe (CePL.L) to the right precuneus (PreCU.R; *p* < 0.01, AlphaSim-corrected). (**B**) Increased effective connectivity from the left cerebellum anterior lobe (CeAL.L) to the right inferior temporal gyrus (ITG.R; *p* < 0.01, AlphaSim-corrected). (**C**) Increased effective connectivity from the right cerebellum posterior lobe (CePL.R) to the left inferior frontal gyrus (IFG.L; *p* < 0.01, AlphaSim-corrected). (**D**) Increased effective connectivity from the left inferior semi-lunar lobule of cerebellum (CeISL.L) to the right superior frontal gyrus (SFG.R; *p* < 0.01, AlphaSim-corrected). (**E**) Summary of the increased effective connectivity and spatial distribution from the cerebellum to the cerebrum.

**Figure 3 brainsci-12-01658-f003:**
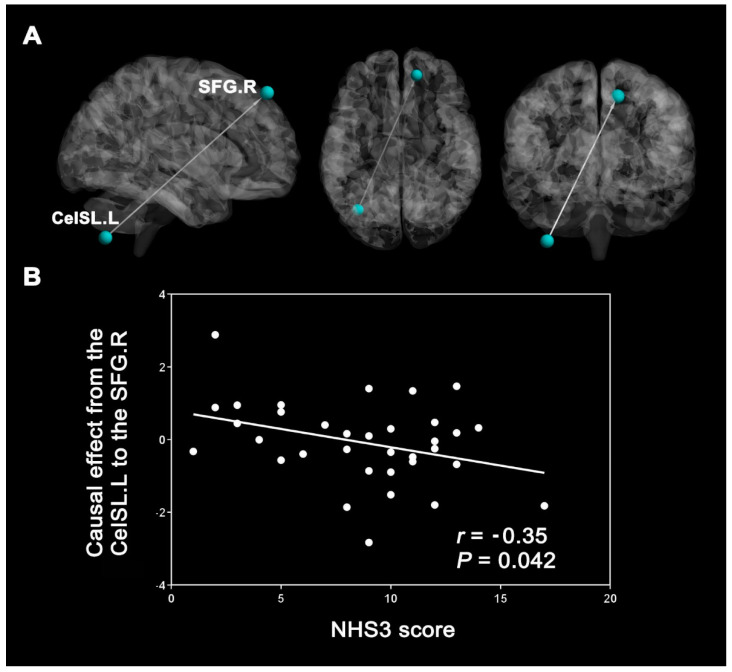
Correlations between the increased effective connectivity (EC) and the National Hospital Seizure Severity Scale (NHS3) scores. (**A**) The EC from the left inferior semi-lunar lobule of the cerebellum (CeISL.L) to the right superior frontal gyrus (SFG.R). (**B**) The NHS3 scores negatively correlated with the increased effective connectivity from the CePL.L to the SFG.R (r = −0.35, *p* = 0.042).

**Table 1 brainsci-12-01658-t001:** Demographic and neuropsychological characteristics of the participants.

Characteristics	JME (n = 34) (Mean ± SD)	HCs (n = 34) (Mean ± SD)	*p* Value
Age (years)	17.38 ± 4.73	19.15 ± 3.46	0.21 ^a^
Handedness (right/left)	34/0	34/0	0.99 ^b^
Sex (males/females)	17/17	11/23	0.14 ^b^
NHS3 score	8.65 ± 3.92	-	-
Mean FD	0.12 ± 0.05	0.11 ± 0.04	0.45 ^a^

^a^ Two-sample *t* test, ^b^ chi-square *t* test; SD, standard deviation; JME, juvenile myoclonic epilepsy; HCs, healthy controls.

**Table 2 brainsci-12-01658-t002:** Brain areas showing significant differences in DC between patients with JME and HCs.

Anatomical Regions	Abbr.	BA	MNI Coordinates			Cluster Size	Maximal *t* Value
			X	Y	Z		
JME >HCs							
Left cerebellum posterior lobe (Cerebellar Tonsil)	CePL.L	-	−3	−51	−48	48	3.54
Right inferior temporal gyrus	ITG.R	20	39	−12	−29	133	5.42
Right superior frontal gyrus	SFG.R	8	15	48	51	64	4.11
JME < HCs							
Left inferior frontal gyrus	IFG.L	47	−45	36	−12	213	−4.47
Left superior temporal gyrus	STG.L	40	−63	−33	15	234	−4.72

*p* < 0.01; AlphaSim-corrected; BA, Brodmann’s area; MNI, Montreal Neurological Institute; CePL.L, Left cerebellum posterior lobe; ITG.R, Right inferior temporal gyrus; SFG.R, Right superior frontal gyrus; IFG.L, Left inferior frontal gyrus; STG.L, Left superior temporal gyrus.

**Table 3 brainsci-12-01658-t003:** Brain areas showing significant differences in EC between patients with JME and HCs.

Anatomical Regions	Abbr.	BA	MNI Coordinates			Cluster Size	Maximal *t* Value
			X	Y	Z		
Seed region: CePL.L							
X→Y							
Right precuneus	PreCU.R	7	21	−63	42	51	3.66
Seed region: ITG.R							
Y→X							
Left cerebellum anterior lobe (Culmen)	CeAL.L	-	−9	−39	−12	39	4.37
Seed region: SFG.R							
Y→X							
left inferior semi-lunar lobule of cerebellum	CeISL.L	-	−36	−69	−54	44	4.16
Seed region: IFG.L							
Y→X							
Right cerebellum posterior lobe (Pyramis)	CePL.R	-	3	−75	−33	43	4.29

*p* < 0.01, AlphaSim-corrected. BA, Brodmann’s area; MNI, Montreal Neurological Institute; PreCU.R, Right precuneus; CeAL.L, Left cerebellum anterior lobe; CePL.R, Right cerebellum posterior lobe; CePL.L, Left cerebellum posterior lobe. CeISL.L, Left inferior semilunar lobule of the cerebellum.

## Data Availability

Not applicable.

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
