# Peer review of "Altered Cerebro-Cerebellar Effective Connectivity in New-Onset Juvenile Myoclonic Epilepsy"

_brainsci, 2022, doi:10.3390/brainsci12121658_

Round 1
Reviewer 1 Report
The manuscript studies the cerebro-cerebellar effective connectivity in new-onset juvenile myoclonic epilepsy through Resting-state fMRI. The study confirms the role of cerbro-cerebellum connections abnormality in epilepsy. Please find the following comments regarding that;
1- The manuscript has a lot of abbreviation that make the following of the text is difficult.
2- Line 23; NHS3 term was used without previous definition
3- Line 311; why did you use ‘’ The encephalic region’’ term rather than cerebrum
4- I cannot find true explanation of the abnormalities in EC among epileptic participants. The authors just mention the abnormality in the area and then its function. Why this change.
5- There in no explanation or discussion of the correlation results
6- As the study provides a new perspective for understanding the cerebro-cerebellar neural 33 circuit - mechanisms in JME. Where are the clinical or research recommendation
7- What is the importance of results
Author Response
非常感谢您对我们稿件的评论,我们的回复在附件中,请参阅附件。

Reviewer 2 Report
In this study, the authors investigate the functional connectivity differences between the patients with JME and the control subjects via degree centrality and effective connectivity analyzes. There is a publication of the authors using the same dataset in 2020. Unlike the current study, the previous one employed a dynamic effective connectivity method and focused on the DMN subdomains detected by the GICA algorithm thus the results of the two studies are not comparable but might be complementary.
The major problem of the current manuscript is that it is very difficult to understand.
Introduction:
41: References to WM abnormality are given([2,3]), but do not fully correspond to the statement.
46: "location of abnormal discharge", "epileptic foci" and "transmission route"; The place of the stated expressions in the subject integrity is not understood.
50: This paragraph should be phrased better.
58: The integrity of the publications in the literature could not be reflected.
Materials and Methods:
121: "lip angle" - "flip angle"
128: (http://rfmri.org/DPARS) - (http://rfmri.org/DPARSF)
130: The sentence needs to be revised
136: <2 mm - is this true?
142: 3×mm 3×mm 3mm3 - please correct.
Results:
Figure 1: it is better to choose the same axial slice positions for A, B, and C.
Table 2: Were all voxels of the statistically significant clusters located within the same anatomical region? Please specify.
There were 5 regions in the results of degree centrality analysis but 4 of them were used as seed regions in effective connectivity analysis. Why was STG.L ignored?
Contrary to what is stated in the manuscript, effective connectivity analysis is not limited to DC analysis results but is applied to all brain voxels. This part needs clarification. It is clearly written in the discussion section only.
Were the voxel distributions of effective connectivity results limited to the reported anatomical regions? Please specify.
It would be better to give the result order for Table 3 in a similar order with the findings in Table 2.
It is better to use an alternative naming for two different areas of the CePL.L in the results of the EC analysis.
Discussion:
The finding of a decrease in the degree centrality of IFG.L and increased effective connectivity with CePL.R should be discussed together.
In the discussion, bilateral SFG was reported which is inconsistent with the findings.
Author Response
Thank you very much for your comments on us. We will write the reply in the attachment. Please see the attachment.

Round 2
Reviewer 1 Report
The authors has answered all the comments except the clinical recommendations. What are the results can clinically recommend?
Author Response
Thanks for reminding us, we have made a simple reply to this question in the part of response.
Our results not only found abnormal DC brain regions in JME patients, but also found abnormal ECs between the cerebellum and the cerebrum based on the seed points from brain regions with different DC values. It is noteworthy that these ECs were all from the cerebellum to the cerebrum. These results suggest that the structure and function of the cerebellum may be affected by the abnormal discharge of epilepsy, or the cerebellum itself may participate in the neural network of epilepsy [1].
Some scholars considered that by stimulating cerebellum to reach thalamic neurons via glutamate fibers, thalamic membrane can be prevented from over-polarization and sudden explosion of action potential [2]. Other study [3], using pharmacological or optogenetic methods, have shown that targeted stimulation of the deep cerebellar nuclei (or glutamate nerve fibers) in laboratory mice can control seizures.
Therefore, we think that the results of our study on the cerebellum can be recommended for clinical, it not only provides potential imaging indicators for the clinical diagnosis of JME, but also provides possible therapeutic targets for non-invasive and targeted anti-epileptic therapy.
- Kim JH, Kim JB, Suh SI. Alteration of cerebello-thalamocortical spontaneous low-frequency oscillations in juvenile myoclonic epilepsy. Acta Neurol Scand 2019;140:252-258.doi:10.1111/ane.13138
- von Krosigk M, Bal T, McCormick DA. Cellular mechanisms of a synchronized oscillation in the thalamus. Science (New York, NY) 1993;261:361-364.doi:10.1126/science.8392750
- Wong JC, Escayg A. Illuminating the Cerebellum as a Potential Target for Treating Epilepsy. Epilepsy currents 2015;15:277-278.doi:10.5698/1535-7511-15.5.277
Reviewer 2 Report
Thanks to the authors.
Only the response to point 8 (3×mm 3×mm 3mm.) is inappropriate. Please correct the notation as follows:
3mm x 3mm x 3mm
Author Response
Thank you very much for pointing out the problem. I am very sorry for my carelessness, and I have revised it in the text.
3mm x 3mm x 3mm